# Is adaptation limited by mutation? A timescale-dependent effect of genetic diversity on the adaptive substitution rate in animals

**Marjolaine Rousselle**[1]*, **Paul Simion**[1,2], **Marie-Ka Tilak**[1], **Emeric Figuet**[1], **Benoit Nabholz**[1], **Nicolas Galtier**[1]

**1** ISEM, Univ. Montpellier, CNRS, EPHE, IRD, Montpellier, France, **2** LEGE, Department of Biology, University of Namur, Namur, Belgium

* marjolaine.rousselle@birc.au.dk

## Abstract

Whether adaptation is limited by the beneficial mutation supply is a long-standing question of evolutionary genetics, which is more generally related to the determination of the adaptive substitution rate and its relationship with species effective population size ($N_e$) and genetic diversity. Empirical evidence reported so far is equivocal, with some but not all studies supporting a higher adaptive substitution rate in large-$N_e$ than in small-$N_e$ species. We gathered coding sequence polymorphism data and estimated the adaptive amino-acid substitution rate $\omega_a$, in 50 species from ten distant groups of animals with markedly different population mutation rate $\theta$. We reveal the existence of a complex, timescale dependent relationship between species adaptive substitution rate and genetic diversity. We find a positive relationship between $\omega_a$ and $\theta$ among closely related species, indicating that adaptation is indeed limited by the mutation supply, but this was only true in relatively low-$\theta$ taxa. In contrast, we uncover no significant correlation between $\omega_a$ and $\theta$ at a larger taxonomic scale, suggesting that the proportion of beneficial mutations scales negatively with species' long-term $N_e$.

**Data Availability Statement:** Data are contained within the manuscript and/or Supporting Information files, and sequence data are deposited

## Author summary

The determinants of the rate at which species adapt to environmental changes are so far poorly understood. In particular, whether adaptation is limited by the mutation supply, which is linked to species population size, is still an open question despite its importance in conservation biology.

Here, we used a comparative population genomic approach to assess the effect of the population mutation supply (approximated by the genetic diversity) on the adaptive substitution rate in animals.

For this we build and analyze a large coding sequence polymorphism dataset covering 50 species from ten diverse groups of animals including insects, molluscs, annelids, birds, and mammals. Thanks to our stratified sampling strategy, which allowed us to compare

under the Bioproject PRJNA530965 in the SRA database.

**Funding:** This work was supported by Agence Nationale de la recherche grant no. ANR-15-CE12-0010 'DarkSideOfRecombination' obtained by NG (https://anr.fr/Project-ANR-15-CE12-0010). The funders had no role in study design, data collection and analysis, decision to publish, or preparation of the manuscript.

**Competing interests:** The authors have declared that no competing interests exist.

closely-related and distantly-related species in a single analysis, we reveal that (i) the supply of beneficial mutations only limits adaptation in low-diversity taxa, such as primates, but not in high-diversity taxa, such as fruit flies, and (ii) low-diversity taxa do not accumulate adaptive substitutions at a slower rate that high diversity taxa, as usually assumed, which may be due to the influence of long-term life history strategies on the proportion of adaptive mutations.

## Introduction

It is widely recognized that adaptation is more efficient in large populations. Firstly, large populations produce a greater number of mutants per generation than small ones, and for this reason are more likely to find the alleles required for adaptation, if missing from the gene pool. Secondly, large populations tend to be genetically more diverse and thus more likely to carry the alleles needed to respond to environmental changes [1]. Lastly, the fixation probability of beneficial mutations is higher in large than in small populations due to the weaker effect of genetic drift in the former. So, whether it be from standing variation or *de novo* mutations, one would expect to observe a higher rate of accumulation of adaptive changes, on average, in large than in small populations [2]. Under a simple population genetic model, in a population of effective size $N_e$, mutations of selection coefficient $s >> 1/N_e$ should accumulate at rate $\sim 4N_e\mu_a s$ if s is small, where $\mu_a$ is the adaptive mutation rate–i.e., the adaptive substitution rate should scale linearly with the population mutation rate $\theta = 4N_e\mu$ (where $\mu$ is the total mutation rate) [3].

This rationale implicitly assumes that the rate of adaptation is limited by the supply of new mutations [4]. It might be, however, that the amount of genetic diversity available in all or most existing populations is sufficient for adaptation, and/or that the ability to adapt to environmental changes is determined in the first place by factors independent from $\theta$, such as the magnitude or frequency of perturbations, the finite set of possible genotypes an organism could reach, or the ability of populations to combine favorable alleles across loci via recombination [5–11]. Finally, this rationale makes the assumption of a constant distribution of the fitness effect of mutations (DFE) across species, whereas it has been suggested that the adaptive mutation rate, $\mu_a$, might be negatively correlated with $N_e$, which further complicates the situation. This is because small populations tend to accumulate deleterious mutations, and the resulting load could offer the opportunity for adaptive, compensatory mutations to arise and spread irrespective of environmental perturbations [10]. Theoretical models can therefore predict a positive, negative, or lack of relationship between the population size and the adaptive substitution rate, depending on the underlying assumptions.

Molecular data offer an unique opportunity to empirically evaluate the correlation between the adaptive substitution rate and $\theta$. More efficient adaptation in high-$\theta$ populations should be reflected by an increased protein evolutionary rate, which can be estimated from coding sequence alignments. The ratio of non-synonymous (i.e. amino-acid changing, dN) to synonymous (i.e. amino-acid conservative, dS) substitution rates, often called $\omega$, is a measure of the protein evolutionary rate that controls for the effects of the divergence time and mutation rate. However, $\omega$ is influenced by adaptation but also by the strength and efficiency of purifying selection against deleterious alleles. To account for this, McDonald and Kreitman (1991, MK) [12] suggested including within-species polymorphism in the analysis. Adaptive mutations are expected to contribute negligibly to the pool of segregating alleles. The ratio of non-synonymous to synonymous polymorphism, therefore, provides an estimator of the expected $\omega$ under

neutrality, i.e., in absence of adaptation, called $\omega_{na}$ (for non-adaptive). Subtracting the neutral expectation $\omega_{na}$ from the observed $\omega$ provides an estimator of the adaptive rate, $\omega_a$, and the proportion of adaptive substitutions, $\alpha$ [13].

Subsequent improvements in the MK method were intended to account for a number of factors that could potentially confound the estimation of $\omega_{na}$, including the prevalence of slightly deleterious segregating alleles and recent demographic effects [14–21]. Improved methods explicitly model the DFE of non-synonymous mutations, while taking information not only from the number of synonymous and non-synonymous single nucleotide polymorphisms (SNPs), but also from the distribution of allele frequencies across SNPs–so-called site frequency spectra (SFS). The $\omega_a$ statistics has a high sampling variance [22] and its estimation can be biased by various factors, such as a fluctuating population size [12,23,24] and GC-biased gene conversion [25–27], implying that MK-based analyses require cautious interpretations.

The first applications of the MK method to large-scale data sets indicated that the adaptive rate is higher in *Drosophila* than in humans [12–14] This is consistent with the prediction of more efficient adaptation in high-$\theta$ populations and with the hypothesis that mutation limits adaptation. These studies were, however, focused on the $\alpha = \omega_a/(\omega_a+\omega_{na})$ statistics, i.e., the proportion of amino-acid substitutions that result from adaptation. $\alpha$ is influenced by $\omega_{na}$ as well as $\omega_a$, and a lower $\alpha$ in humans than in *Drosophila* might mainly reflect a higher rate of non-adaptive amino-acid substitution in the former. Indeed, purifying selection against deleterious mutations is likely less effective in small populations due to increased genetic drift [28]. Comparative studies focused on $\omega_a$ have only revealed tenuous positive effects of $\theta$ on the adaptive rate in mammals, flies and plants [29–31]. The largest scale analysis of this sort used 44 pairs of non-model species of animals occupying a wide range of $\theta$ [18]. This latter study reported a significantly positive relationship between $\theta$-related life history traits and $\alpha$, consistent with previous literature, but this was entirely due to the non-adaptive component. Galtier [18] failed to detect any effect of $\theta$ on $\omega_a$, despite using various models for the distribution of fitness effects and accounting for a number of potential confounding factors. This result did not support the hypothesis that adaptation is limited by the population mutation rate.

So, the evidence so far regarding the relationship between the adaptive substitution rate and the population mutation rate is equivocal. Existing comparative studies have involved distinct methodological approaches, both in terms of species sampling and adaptive substitution rate estimation. In particular, these studies were conducted at different evolutionary scales, which might partially explain their somewhat discordant results. In the short term, an increase in $N_e$ is expected to boost the adaptive substitution rate if the mutation supply is limiting. In the long run, differences in $N_e$ could also lead to changes in the DFE, and particularly in the proportion of beneficial mutations, due to the fact that small-$N_e$ species may be pulled away from their fitness optimum via genetic drift [7,18,32]. How these two opposing forces interact and combine to determine the relationship between $\omega_a$ and $\theta$ is still unknown, in the absence of a multi-scale study.

In this study, we test the effects of the evolutionary time-scale on the relationship between the adaptive substitution rate ($\omega_a$) and the population mutation rate ($\theta$). We gathered coding sequence polymorphism data in 4–6 species from each of ten distant groups of animals with markedly different $\theta$. Our results reveal that the relationship between $\omega_a$ and $\theta$ varies depending on the considered taxonomic scale, i.e. depending on whether we compare closely related species or distantly related taxa. We report a positive relationship between $\omega_a$ and $\theta$ within groups, and the strength of this relationship weakens as $\theta$ increases, indicating that adaptation is limited by beneficial mutations in small-$\theta$ animal species. At a larger taxonomic scale, in contrast, we find a weak negative correlation between $\omega_a$ and $\theta$, with, for instance, primates and ants showing a higher adaptive substitution rate than mussels and fruit flies. This is in line

with the hypothesis that long-term $N_e$ influences the DFE, and particularly the proportion of adaptive mutations.

## Results

### Data sets

We assembled a data set of coding sequence polymorphism in 50 species from ten taxonomic groups, each group including 4 to 6 closely-related species (S1 Table). The ten taxa we analyzed were Catarrhni (Mammalia, hereafter called "primates"), Passeriformes (Aves, hereafter called "passerines"), Galloanserae (Aves, hereafter called "fowls"), Muroidea (Mammalia, hereafter called "rodents"), Lumbricidae (Annelida, hereafter called "earth worms"), *Lineus* (Nemertea, hereafter called "ribbon worms"), *Mytilus* (Mollusca, hereafter called "mussels"), Satyrini (Lepidoptera, hereafter called "butterflies"), *Formica* (Hymenoptera, hereafter called "ants"), and *Drosophila* (hereafter called "flies").

Data for five groups (primates, passerines, fowls, rodents and flies) were obtained from public databases. Data for the other five groups were newly generated via exon capture in a total of 242 individuals from 22 species (Table 1) and we obtained sufficient data for 216 of them (~89%). The average coverage was of 9X in ants, 23X in butterflies, 10X in earth worms, 28X in ribbon worms and 26X in mussels (average of median coverage per species). The percentage of targeted coding sequences for which at least one contig was recovered ranged from 31.9% (for *Lumbricus terrestris*, the species with the maximal divergence from the species used to design the baits) to 88.2% across species (median = 78.8%, Table 1).

We assessed contamination between samples from distinct species using CroCo [33]. Overall, the inter-groups connection in S1 Fig indicates a low level of cross-contamination: when there were connections between taxonomic groups, on average they concerned 38 contigs

Table 1. Summary of the number of targeted transcripts recovered in the capture experiment.

| Species | Group | Targeted transcripts | Recovered transcripts | Percentage of recovered among targeted transcripts |
|---|---|---|---|---|
| *Formica fusca* | ants | 1810 | 1427 | 78.8 |
| *Formica sanguinea* | ants | 1810 | 1396 | 77.1 |
| *Formica pratensis* | ants | 1810 | 1398 | 77.2 |
| *Formica cunicularia* | ants | 1810 | 1406 | 77.7 |
| *Maniola jurtina* | butterflies | 2235 | 1921 | 86.0 |
| *Melanargia galathea* | butterflies | 2235 | 1713 | 76.6 |
| *Pyronia tithonus* | butterflies | 2235 | 1823 | 81.6 |
| *Pyronia bathseba* | butterflies | 2235 | 1864 | 83.4 |
| *Aphantopus hyperanthus* | butterflies | 2235 | 1772 | 79.3 |
| *Allolobophora chlorotica L1* | earth worms | 2955 | 2293 | 77.6 |
| *Allolobophora chlorotica L2* | earth worms | 2955 | 2315 | 78.3 |
| *Allolobophora chlorotica L4* | earth worms | 2955 | 1732 | 58.6 |
| *Aporrectodea icterica* | earth worms | 2955 | 2321 | 78.5 |
| *Lumbricus terrestris* | earth worms | 2955 | 943 | 31.9 |
| *Lineus sanguineus* | ribbon worms | 1725 | 1251 | 72.5 |
| *Lineus ruber* | ribbon worms | 1725 | 1521 | 88.2 |
| *Lineus lacteus* | ribbon worms | 1725 | 1516 | 87.9 |
| *Lineus longissimus* | ribbon worms | 1725 | 1505 | 87.2 |
| *Mytilus galloprovincialis* | mussels | 2181 | 1820 | 83.4 |
| *Mytilus edulis* | mussels | 2181 | 1721 | 78.9 |
| *Mytilus trossulus* | mussels | 2181 | 1740 | 79.8 |
| *Mytilus californianus* | mussels | 2181 | 1808 | 82.9 |

identified as contaminants, with the worst case being the 172 contigs identified as contaminants between the assembly of *Lineus sanguineus* and *Mytilus galloprovincialis*. Connections between assemblies from closely related species were very likely false positive cases, especially since the intensity of the within-group connections was congruent with the phylogenetic distance between species within taxa. Regardless, all contigs identified as potential contaminants were excluded from the dataset in downstream analyzes as a cautionary measure.

Within each group, we focused on orthologous contigs (**S2 Table**), predicted open reading frames, and called the diploid genotypes of individuals for every coding position. The SNPs counts obtained after genotyping are summarized in **S3 Table**. We obtain less than a thousand SNPs in only two species, the minimum being 153 for *Lineus longissimus*, in which we were only able to recover data for six individuals. We recovered an average of 8,459 SNPs per species in ants, 7,950 in butterflies, 4,763 in earth worms, 8,347 in ribbon worms, 19,750 in mussels, 10,191 in primates, 25,534 in rodents, 40,870 in passerines, 8,488 in fowls and 195,398 in flies.

In conclusion, the capture experiment seems suitable for recovering population coding sequence data for several closely related species—here, the maximum divergence between species within a taxonomic group was 0.2 subst./site, i.e. the divergence between *Lumbricus terrestris* and *Allolobophora chlorotica L1*.

## Between-groups relationship between the population mutation rate (θ) and the adaptive substitution rate ($\omega_a$)

We used Galtier's (2016) version of the MK method [18] introduced by Eyre-Walker and Keightley (2009) [16], accounting for the effect of slightly beneficial non-synonymous mutations (see Methods). Two strategies were adopted to combine SFS information from distinct species in a group-level estimate of $\omega_a$, thus accounting for the problem of phylogenetic non-independence between species. For both strategies, we first calculated the dN/dS ratio ω at the group-level, i.e., by averaging across all branches of the tree (see Methods). Our first estimator, which we called $\omega_{a[P]}$, was obtained by pooling SFS from distinct species within a group, separately for synonymous and non-synonymous SNPs (as in [34]), before fitting the model and estimating the parameters. This estimate combines data across species weighting each species equally, thus alleviating the effect of species-specific demographic history.

We then computed the relationship between $\omega_{a[P]}$ estimates and the across-species average nucleotide diversity, $\pi_s$, which was taken as an estimate of θ. We did not detect a significant positive relationship between $\omega_{a[P]}$ and the across-species average nucleotide diversity, $\pi_s$, taken as an estimate of θ, but the estimates based on all mutations rather suggest a weak negative relationship (**Fig 1A**).

Recent studies in birds and more recently primates indicated that GC-biased gene conversion (gBGC) may lead to overestimation [25,26] or underestimation of $\omega_a$ [27]. Interestingly, gBGC does not affect genomic evolution with the same intensity in all organisms [35]. To avoid bias in the estimation in species where gBGC is active, we restricted the SNP and substitution data to GC-conservative changes, which are not influenced by gBGC. We found a non-significant positive correlation $\omega_{a[P]GC\text{-conservative}}$ and θ (**Fig 1C**).

Our second estimator of the adaptive rate at the group level, which we called $\omega_{a[A]}$, was obtained by calculating the across-species arithmetic mean of $\omega_{na}$ within a group, and by then subtracting this average from ω. We suggest that $\omega_{a[A]}$ is a reasonable estimator of the adaptive rate with fluctuating population size if the pace of fluctuations is sufficiently slow, such that the sampled species have reached the selection/drift equilibrium (Supplementary Material **S1 Text**). The use of this estimate seems to confirm the absence of a positive relationship between $\omega_{a[A]}$ and $\omega_{a[A]GC\text{-conservative}}$ and $\pi_s$, but rather suggest a weak negative relationship (**Fig 1B and 1D**).

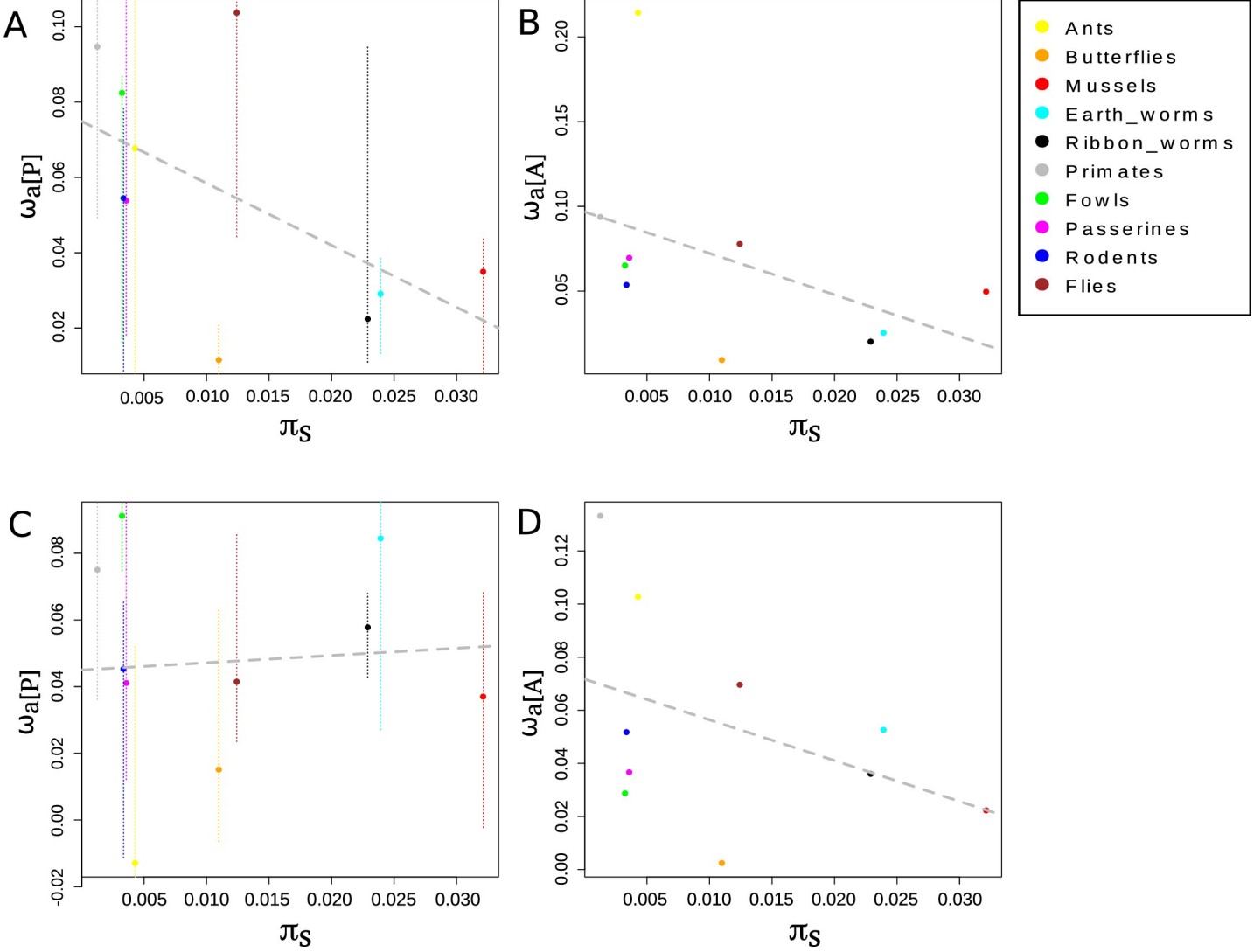

**Fig 1. Relationship between group-level $\omega_a$ and group-level $\pi_s$.** A: $\omega_a$ was estimated by pooling SFS across species within a group ($\omega_{a[P]}$) using all mutations. B: $\omega_a$ was estimated by the averaging of $\omega_{na}$ across species within a group ($\omega_{a[A]}$) using all mutations. C: $\omega_a$ was estimated by pooling SFS across species within a group ($\omega_{a[P]}$) using only GC-conservative mutations. D: $\omega_a$ was estimated by the averaging of $\omega_{na}$ across species within a group ($\omega_{a[A]}$) using only GC-conservative mutations. Group level $\pi_s$ was estimated by averaging species-level $\pi_s$ across closely related species. Black dotted lines represent the regression line when the Spearman correlation is significant and grey dotted lines when the Spearman correlation is non-significant. Thin vertical dotted lines represent the 95% confidence intervals obtained by bootstrapping SNPs.

### Relationship between life history traits and $\omega_a$

In view of the absence of a positive relationship between $\omega_a$ and the population mutation rate at a large taxonomic scale, we tested life history traits as other potential explanatory variables of $\omega_a$. It has previously been shown that at a large taxonomic scale, genetic diversity is accurately predicted by some life history traits, with long-lived or low-fecundity species being genetically less diverse than short-lived or highly fecund ones [36]. This is usually interpreted as life history traits being representative of the long-term population size.

In our data set, all $\log_{10}$ transformed life history traits but adult size and longevity were correlated with $\pi_s$ (propagule size: regression test $r^2 = 0.58$, p-value = 2.4e-10, adult size: regression test $r^2 = 0.04$, p-value = 0.10, longevity: regression test $r^2 = 0.30$, p-value = 0.14, body mass: regression test $r^2 = 0.095$, p-value = 0.03, fecundity: regression test $r^2 = 0.61$, p-

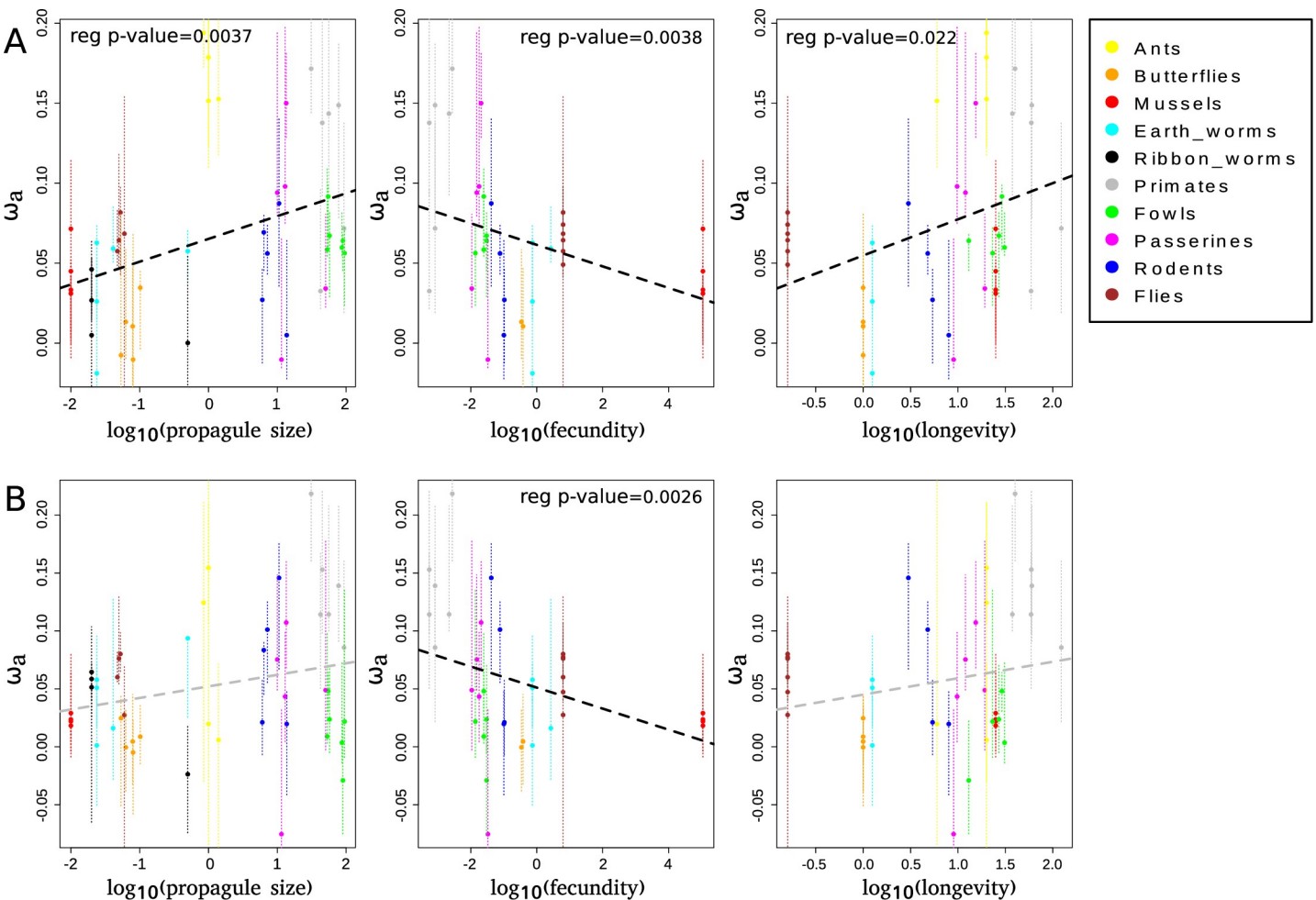

**Fig 2. Relationship between species-level $\omega_a$ and life history traits.** A: $\omega_a$ is estimated using all mutations. B: $\omega_a$ is estimated using only GC-conservative mutations. Black dotted lines represent significant regressions across taxonomic groups and grey dotted lines non-significant ones. Thin vertical dotted lines represent the 95% confidence intervals obtained by bootstrapping SNPs.

value = 8.9e-9). When estimating the per-group $\omega_a$, we found significant negative relationships between $\omega_{a[P]GC\text{-}conservative}$ and adult size (Spearman correlation coefficient = 0.65,p-value = 0.049) (**S2B Fig**), but did not otherwise found significant relationships with life history traits. However the signs of the correlation coefficients were indicative of a negative relationship between those life history traits and both $\omega_a$ and $\omega_{a[GC\text{-}conservative]}$ (**S2** and **S3** Figs).

When considering all 50 species (i.e. without controlling for phylogenetic inertia) and all mutations, we found a negative relationship between $\omega_a$ and $\log_{10}$ transformed fecundity (regression test, $r^2$ = 0.094, p-value = 0.038), as well as a positive relationship with $\log_{10}$ transformed longevity (regression test, $r^2$ = 0.10, p-value = 0.022) and $\log_{10}$ transformed propagule size (regression test, $r^2$ = 0.13, p-value = 0.0073) (**Fig 2A**). When using only GC-conservative mutations, the relationships were similar but only significant with fecundity (regression test, $r^2$ = 0.11, p-value = 0.026) (**Fig 2B**).

We also found a significant negative relationship between $\omega_{na}$ and fecundity, and positive relationships between $\omega_{na}$ and propagule size, body mass, propagule size and longevity (**S4A Fig**). This remained true when using only GC-conservative mutations (**S4B Fig**).

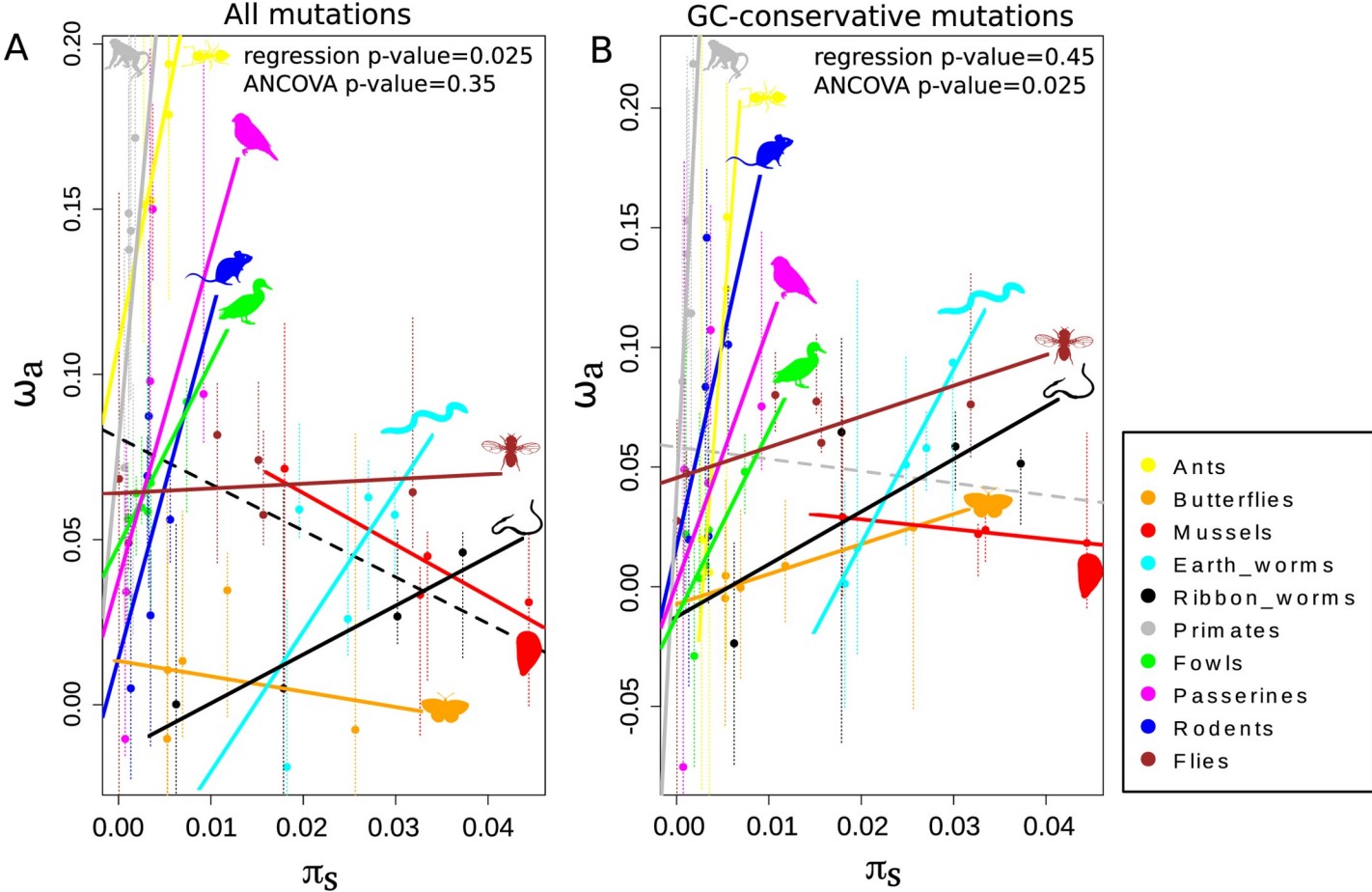

**Fig 3. Relationship between species-level $\omega_a$ and $\pi_s$.** A: $\omega_a$ is estimated using all mutations. B: $\omega_a$ is estimated using only GC-conservative mutations. Black dotted lines represent significant regressions across taxonomic groups and grey dotted lines non-significant ones. Thin vertical dotted lines represent the 95% confidence intervals obtained by bootstrapping SNPs.

### Within-group relationship between θ and $\omega_a$

To assess the within-group effect of $\pi_s$ on $\omega_a$, we performed an analysis of covariance (ANCOVA) with the taxonomic group as a categorical independent variable, as in [29]. The principle of this analysis is to fit a set of parallel lines (one for each taxonomic group) and test whether their common slope is significantly different from zero. Moreover, we tested if the relationship between $\omega_a$ and $\pi_s$ or life history traits differs between taxonomic groups by testing whether the lines have different intercepts.

By this strategy, we found that $\omega_a$ and both $\pi_s$ and $\log_{10}(\pi_s)$ were significantly positively correlated when using only GC-conservative mutations (ANCOVA p-value = 2.8e-02 and 3.1e-03, respectively) (**Fig 3B**). $\omega_a$ was only marginally positively correlated with $\log_{10}(\pi_s)$ when using all mutations (ANCOVA p-value = 7.6e-02). We also found that there was a significant variation between the intercepts (ANCOVA p-value<1e-03), as well as a significant interaction between the dependent variable and the categorical independent variable (ANOVA p-value = 1.6e-02) when using only GC-conservative mutations.

Those results support the existence of a positive relationship between $\omega_a$ and θ within groups, with the slope of the relationship differing between groups. This is consistent with the hypothesis that within a group, higher-θ species are more likely to find and fix adaptive

substitutions than low-θ species, in line with the hypothesis that mutation limits adaptation. **Fig 3** shows that the slopes of the within-group $\omega_a/\theta$ correlations decreased with group-level $\pi_s$, and we actually found a significant negative correlation between these two quantities both when using all or only GC-conservative mutations (Spearman correlation coefficient = -0.77, p-value = 1.4e-02). This interestingly suggests that the limitation of adaptation by the supply of adaptive mutations is effective and strong in small-θ groups (e.g. primates, rodents, ants), but not in high-θ groups of animals (e.g. flies, mussels, butterflies), where the $\omega_a/\theta$ relationship is essentially flat (**Fig 3**).

When analyzing the per-species non-adaptive substitution rate, we found a global negative relationship between $\omega_{na}$ and $\pi_s$ (using both all mutations and only GC-conservative mutations: regression test $r^2$ = 0.16, p-value = 0.0022 and $r^2$ = 0.33, p-value = 7.7e-6, respectively) (**S4A and S4B Fig**), and a significantly negative relationship within groups (ANCOVA p-value = 1.9e-02 and p-value = 1.8e-03, respectively). This was consistent with the expectations of the nearly neutral theory of evolution [28], and with previous empirical results [18,37]. The estimated ratio of adaptive to total non-synonymous substitutions, α, behaved more or less similarly to $\omega_a$ (**S5 Fig**).

## Control for fluctuations in $N_e$

We were concerned that the positive correlation between $\omega_a$ and $\pi_s$ might have been due to an artifact generated by past fluctuations in population size. Such fluctuations violate the assumption that the regime of selection/drift has been constant over the considered time period. This has been shown to yield spurious evidence of positive selection, and possibly a spurious positive correlation between $\omega_a$ and $\pi_s$ [23,24]. To test this, we simulated coding sequence evolution under several demographic scenarios with four regimes of demographic fluctuations, with a three or thirty-fold ratio between the low and high $N_e$, and a high or low long-term $N_e$ (see Material and Methods and **S6 Fig**). We found that the only scenario where demographic fluctuations could lead to a detectable positive correlation between $\omega_a$ and $\pi_s$ was that with the highest long-term $N_e$ and highest difference between the low and high $N_e$ (see **S7B Fig**, regression test $r^2$ = 0.07, p-value = 9.5e-03). The correlation disappeared when we used a ten-fold smaller long-term $N_e$, whereas we empirically observed that the correlation between $\omega_a$ and $\pi_s$ was stronger for small long-term $N_e$ groups (**Fig 2**). These simulations therefore suggested that ancient demographic fluctuations could not explain our finding of a positive within-group correlation between $\omega_a$ and $\pi_s$ in low-θ groups. We also estimated the $F_{is}$ statistics in all species, where $F_{is}$ measures the inbreeding coefficient of an individual relative to the subpopulation, to check that a potential population substructure would not influence the estimations of statistics based on polymorphism data. We found no significant correlation between $F_{is}$ and $\omega_a$ (regression test p-value = 5.9e-01) or $\pi_s$ (p-value = 2.9e-01).

## Discussion

### Influence of θ on $\omega_a$: A two-scales mechanism

In this study, we analyzed a 50-species population genomic data set to assess the relationship between the adaptive substitution rate and the population mutation rate and test the hypothesis that mutation limits adaptation in natural populations of animals.

We found that the relationship between $\omega_a$ and θ depended on the considered timescale, which is expected if the assumption of a fixed DFE across divergent taxa does not hold. At a recent evolutionary scale (i.e., neutral divergence <0.2 subst./site), we found a significant positive correlation between $\omega_a$ and $\pi_s$ (**Fig 3**). Interestingly, the slope of the relationship differed significantly among taxonomic groups, and this slope itself was negatively correlated with the

group average $\pi_s$. Otherwise, estimates at the group level revealed a weak but consistently negative relationship between $\omega_a$ and $\pi_s$, and between $\omega_a$ and various life history traits correlated with the long-term $N_e$ (**Figs 1** and **3**). This time scale-dependent behavior of the $\omega_a/\theta$ relationship was here demonstrated via the analysis of a single, multi-scale dataset, somehow reconciling earlier taxon-specific studies on the subject [4,9,18,29–31,38].

## Relationship between $\theta$ and $\omega_a$—A real causative link or an artifact?

Our ANCOVA analysis revealed that the slopes of the relationships between $\omega_a$ and $\pi_s$ within each taxonomic group were significantly different from zero, demonstrating the existence of a positive link between $\omega_a$ and $\pi_s$ within groups (**Fig 2**). We were concerned that this relationship may have resulted from a bias in the MK approach, instead of being a true biological signal. Indeed, the MK approach implicitly assumes that the regime of selection/drift has been constant over the considered time period, i.e. since the divergence between the focal and outgroup species. If however the selection/drift regime had changed (e.g. via a change in effective population size) between the period during which divergence had accumulated and the period during which polymorphism was built, this could lead to overestimation or underestimation of $\omega_a$ [23,24]. Here, we used the so-called $r_i$'s nuisance parameters [39] to control for recent changes in $N_e$.

In contrast, ancient $N_e$ changes that affect coding sequence divergence are virtually impossible to trace. We showed in a previous simulation-based study that ancient demographic fluctuations could lead to severely overestimated $\alpha$ and $\omega_a$—an upward bias which is exacerbated when the true adaptive substitution rate is low [23]. Moreover, it has been shown by modeling single changes in $N_e$ that in the presence of slightly deleterious mutations, an increase in $N_e$ in the past could yield spurious evidence of positive selection, which can lead to a spurious positive correlation between $\omega_a$ and $\pi_s$ [24].

We used simulations to test if demographic fluctuations could lead to such a correlation. Our results suggested that long-term fluctuations were not responsible for the positive link between $\omega_a$ and $\pi_s$ that we report. In addition, the gradual decrease in the slope of the relationship with per-group average $\pi_s$ was also consistent with the fact that the relation is genuine, because (i) we do not expect the demographic fluctuation regime to correlate with the average $\pi_s$ of the group, and (ii) there was no relationship between the inter-group variation in $\pi_s$ and the average $\pi_s$ of the group (Spearman correlation test: p-value = 4.7e-01).

A recently developed method allows the estimation of $\alpha$ and $\omega_a$ using polymorphism data alone [20], thus avoiding the assumption of time constancy of the drift/selection regime. However, estimates of $\alpha$ and $\omega_a$ by this method deserve a specific interpretation, as they represent the rate of adaptive evolution of the species during its very recent history, and not the one of its long-term history. This method requires high quality datasets and highly polymorphic species, and it was not applicable to our dataset, in which species and groups differ widely in terms of SNP numbers (**S3 Table**).

## Positive relationship between $\omega_a$ and $\pi_s$ among closely-related species

Our findings are therefore consistent with the existence of a genuine link between the adaptive substitution rate and $\theta$, which would support the hypothesis that, in several groups of animals, the rate of adaptation is limited by the supply of beneficial mutations. The slope of the relationship was particularly steep in ants, fowls, passerines, rodents and primates (**Fig 2**). For instance, the estimated adaptive rate in rhesus macaque (*Macaca mulatta*: $\pi_s = 0.0018$) was more than 3-fold higher than that of humans (*Homo sapiens*: $\pi_s = 0.0006$). Note that this interpretation relies on the assumption that different species from a given taxonomic group share the same DFE and, in particular, the same proportion of beneficial mutations. Castellano et al.

[40] compared the DFE across closely related species (great apes) and found that the deleterious DFE is quite stable across great apes, comforting us in our assumption that the DFE is expected to be similar between closely related species but different between distantly related species.

Our results are also consistent with previous analyses of the relationship between $\omega_a$ and $\pi_s$ at a relatively recent time scale [29]. Finally, it is consistent with the finding that strong selective sweeps are more abundant in species of great apes with a large population size [4].

Interestingly, we found that the relationship between $\omega_a$ and $\pi_s$ was significantly stronger in low-diversity than high-diversity groups. In flies, a high-diversity group, the slope of the linear regression between the two variables was only 1.3, whereas it was between 7.8 and 77 in the four vertebrate groups. In mussels, i.e. the taxonomic group with the highest average diversity in our dataset, we detected no significant relationship between $\omega_a$ and $\pi_s$, with the slope being very close to zero (-0.4). It is possible that in such organisms the adaptive evolutionary rate is not limited by the mutation supply: the standing variation and/or the influx of new mutations are sufficient for proteins to find the required alleles. This is consistent with the results of [9], that showed that patterns of adaptation to insecticides in natural *Drosophila melanogaster* populations are incompatible with the hypothesis that adaptation is mutation-limited. This is also consistent with the results of Jensen and Bachtrog [41], who found very similar rates of adaptation between two *Drosophila* species with different $N_e$.

Finally, the results shown in **Fig 3** corroborate theoretical predictions indicating that when $\theta$ is sufficiently large, it is the species ability to combine beneficial alleles across loci that limits adaption rather than the strength of selection or the mutation supply [10]. Our results suggest that this situation applies to high-$\theta$ groups of animals, such as *Drosophila*, but not to small-$\theta$ ones, such as primates. Indeed, one should keep in mind that the two variables we analyze here, $\pi_s$ and $\omega_a$, are potentially affected by the effects of interference between segregating mutations [17]. Weissman & Barton [10], following Gillespie [42], explicitly modeled linkage between beneficial mutations and showed that the effect of $N_e$ on the adaptive rate is expected to saturate when $N_e$ is sufficiently large. The neutral genetic diversity is also expected to be affected by linked selection [43,44], to an extent that still deserves to be properly assessed [44]. Quantifying the effect of linked selection on the neutral and selected variation, and its relationship with $N_e$, is a current challenge and would help interpreting results such as the ones we report here.

In the above, we interpret the detected relationship between $\omega_a$ and $\pi_s$ in terms of mutation-limited adaptation. It should be noted, however, that $\pi_s$ is only an indirect proxy for the supply of beneficial mutations. In particular, the expected population frequency of large-effect deleterious mutation is essentially independent of $N_e$ and $\pi_s$. So if adaptation most often involved preexisting, large effect mutations that shift from deleterious to beneficial as the environment changes, then our results would probably require a different explanation [45]. Another important caveat is that the effective population size relevant tot the neutral genetic diversity may differ from the effective population size generating beneficial mutations [9]. This is because $\pi_s$ is influenced by ancient bottlenecks and selective sweeps, i.e. it is influenced by the long-term $N_e$, whereas the ability for a population to *de novo* find the required beneficial mutation after an environmental change depends on the recent, contemporary $N_e$. More knowledge about the effect size of mutation that contribute to adaptation and the balance between standing variation and *de novo* mutations, would therefore appear needed for an enhanced interpretation of our results.

## What are the determinants of $\omega_a$ across distantly related taxa?

We used two approaches to estimate the adaptive substitution rate at the group level. Both supported a negative among-group relationship between $\omega_a$ and $\pi_s$, and between $\omega_a$ and life

history traits that have been shown to be linked to the long-term effective population size [36] (**Fig 1**, **S2 Fig**, **S3 Fig** and **Fig 3**). As different sets of genes were used in the different groups of animals, the gene content might have influenced our results. Indeed, Enard et al. [46] showed that genes interacting with viruses experience a significantly higher adaptive substitution rate, thus demonstrating the importance of the gene sampling strategy in comparative studies. In the exon capture experiment, a subset of genes was randomly sampled from an existing transcriptome reference, whereas all available genes were used in the other species (provided that they were present in all species within a group). We do not see any particular reason why the gene sample would be biased with respect to virus interacting proteins in some specific groups, and we did not detect any effect of data type (i.e. exon capture *vs*. genome-wide) on $\omega_a$. Our results may also be influenced by differences between groups in terms of selection on codon usage. It has been shown that in *Drosophila* synonymous mutations are subject to both weak and strong selection [47, 48], which in turn has been shown to potentially lead to an upward bias in the estimation of $\alpha$ [49] (at least when $\alpha$ is estimated as $\left(1 - \frac{d_S}{d_N} * \frac{p_n}{p_s+1}\right)$. On the contrary, there is no evidence for effective translational selection on codon usage in small-$\theta$ species [35]. As such, the slightly negative among-group relationship we report may actually be weakened by the fact that in high-$\theta$ species, $\pi_s$ is underestimated and $\omega_a$ is overestimated due to selection on codon usage, but this remains to be tested more formally.

Our results are consistent with the results of Galtier [18], who analyzed the relationship between $\omega_a$ and $\pi_s$ in a transcriptomic dataset of 44 distantly related species of animals. Indeed, the main analysis in Galtier [18] revealed no significant correlation between $\omega_a$ and $\pi_s$, but various control analyses (particularly using GC or expression restricted datasets) yielded a significantly negative correlation between the two variables. This suggests that the mutation limitation hypothesis does not accurately account for the variation of $\omega_a$ at a large taxonomic scale, implying that factors other than $\theta$ must be at work here.

First, such potential factor could be related to genome structure, and in particular to the compactness of genome that would influence the strength of Hill-Robertson interference [50]: assuming that large $\theta$ species have more compact genomes, they could be more impacted by interference between nearby adaptive mutations, which would potentially decrease their rate of adaptation.

Second, it should be recalled that the expected adaptive substitution rate is in large part determined by the rate of environmental change [7,51]. If one assumes that species with a longer generation time undergo a higher per generation rate of environmental change, then we would expect a higher adaptive substitution rate in long-lived species (typically species with small long-term population size). This is consistent with our observation that $\omega_a$ is positively correlated to longevity [36].

Lourenço et al. [7] simulated protein evolution under Fisher's geometrical model (FGM) and reported that the adaptive substitution rate is an increasing function of the dimensionality of the phenotypic space, which is as a representation of of the complexity of the evolving phenotype. This is because the probability that a new mutation is in the optimal direction decreases as the number of potential directions increases, such that the average adaptive walk takes more steps in a high-dimension than a low-dimension space [7,52]. Complexity *sensus* FGM is hard to quantify in a biologically relevant way. To argue that primates and birds are more complex than mussels and worms does not seem particularly relevant when considering the organism level. Different measures of complexity have been considered at the molecular or cellular level, such as genome size, gene or protein number, number of protein-protein interactions, number of cell types. These seem to point towards a higher complexity in mammals than insects, for instance [37,38], consistent with the idea of a greater genomic complexity of

species with smaller $N_e$. Fernández and Lynch [53] suggested that the accumulation of mildly deleterious mutations in small populations induces secondary selection for protein–protein interactions that stabilize key gene functions, thus introducing a plausible mechanism for the emergence of molecular complexity [53]. If the number of protein-protein interactions is a relevant measure of proteome complexity, then this might contribute to explain our findings of a higher adaptive substitution rate in low-θ than in high-θ groups.

Finally, Huber et al. [32] suggested that variations in the adaptive mutation rate across distantly related taxa could be modulated by the long-term $N_e$ via the mean distance of the population to the fitness optimum. According to this hypothesis, groups of species having evolved under small $N_e$ in the long run would be further away from their optimum, compared to larger-$N_e$ groups, due to an increased rate of fixation of deleterious mutations, and for this reason would undergo a larger proportion of beneficial, compensatory mutations. Empirical analyses of SFS based on large samples are consistent with the hypothesis that humans are on average more distant to their optimum than flies [32].

To sum up, our results suggest that factors linked to species long-term effective population size affect the DFE, i.e., the proportion and rate of beneficial mutation would be non-independent of the long-term $N_e$. We suggest that the proteome is probably more complex and further away from its optimal state in small-$N_e$ than in large-$N_e$ groups of animals, which might contribute to increasing the steady-state adaptive rate in the former, thus masking the effect of mutation limitation in across-group comparisons.

## Conclusion

In this study, we sampled a large variety of animals species and demonstrated a timescale-dependent relationship between the adaptive substitution rate and the population mutation rate, that reconciles previous studies that were conducted at different taxonomic scales. We demonstrate that the relationship between the adaptive substitution rate and θ within closely related species sharing a similar DFE is shaped by the limited beneficial mutation supply, whereas the between-group pattern probably reflects the influence of long-term population size on the proportion of beneficial mutations. Our results provide empirical evidence for mutation-limited adaptive rate at whole proteome level in small-$N_e$ groups of animals, while stressing the fact that DFE is not independent of the long-term effective population size–a crucial factor that must be properly accounted for in large-scale comparative population genomic analyses.

That adaptation is only mutation-limited in low-θ taxa, if confirmed, has implications in conservation biology. Our results suggest that enhancing the genetic diversity of endangered taxa by promoting gene flow between disconnected populations (genetic rescue) is indeed likely to increase the chances of survival by adaptation in low-θ groups of animals, such as mammals and birds, but probably not in high-θ taxa, such as butterflies and marine mollusks for instance. Along the same lines, our results would predict the existence of a negative relationship between θ and species extinction rate in small-$N_e$ but not in large-$N_e$ taxa, a prediction that could be tested via the analysis of diversification patterns across phylogenies.

## Material & methods

### Data set

Genomic, exomic and transcriptomic data from primates, passerines, fowls, rodents and flies were retrieved from the SRA database. Detailed referenced, bioprojects and sample sizes are provided in **S1 Table**. The minimal sample size was five diploid individuals (in *Papio anubis*) and the maximum was 20 (in seven species).

Exon capture data were newly generated in ants, butterflies, mussels, earth worms and ribbon worms. We gathered tissue samples or DNA samples for at least eight individuals per species and four or five species per group. Reference transcriptomes were obtained from previously published RNA-seq data in one species per taxonomic group [36,54,55]. Details of the species and numbers of individuals are presented in **S1 Table**.

## Multiplexed target capture experiment

DNA from whole animal body (ants), body section (earth worms, ribbon worms), mantle (mussels) or head/thorax (butterflies) was extracted using DNAeasy Blood and Tissue kit (QIAGEN) following the manufacturer instructions. About 3 μg of total genomic DNA were sheared for 20 mn using an ultrasonic cleaning unit (Elmasonic One). Illumina libraries were constructed for all samples following the standard protocol involving blunt-end repair, adapter ligation, and adapter fill-in steps as developed by [56] and adapted in [57].

To perform target capture, we randomly chose contigs in five published reference transcriptomes (*Maniola jurtina* for butterflies [54], *Lineus longissimus* for ribbon worms [36], *Mytilus galloprovincialis* for mussels [36], *Allolobophora chlorotica L1* for earth worms [36], and *Formica cunicularia* for ants [55]) in order to reach 2Mb of total sequence length per taxon (~2000 contigs). 100nt-long baits corresponding to these sequences were synthesized by MYbaits (Ann Arbor, MI, USA), with an average cover of 3X.

We then performed multiplexed target capture following the MYbaits targeted enrichment protocol: about 5 ng of each library were PCR-dual-indexed using Taq Phusion (Phusion High-Fidelity DNA Polymerase Thermo Scientific) or KAPA HiFi (2× KAPA HiFi HotStart ReadyMix KAPABIOSYSTEMS) polymerases. We used primers developed in [58]. Indexed libraries were purified using AMPure (Agencourt) with a ratio of 1.6, quantified with Nanodrop ND-800, and pooled in equimolar ratio. We had a total of 96 combinations of indexes, and two Illumina lanes, for a total of 244 individuals. This means that we had to index two (rarely three) individuals with the same combination to be sequenced in the same line. When this was necessary, we assigned the same tag to individuals from distantly related species (i.e. from different groups). Exon capture was achieved according to the Mybaits targeted enrichment protocol, adjusting the hybridization temperature to the phylogenetic distance between the processed library and the baits. For libraries corresponding to individuals from the species used to design baits, we used a temperature of 65˚C during 22 h. For the other ones we ran the hybridization reactions for 16 h at 65˚C, 2 h at 63˚C, 2 h at 61˚C and 2 h at 59˚C. Following hybridization, the reactions were cleaned according to the kit protocol with 200 μL of wash buffers, and hot washes were performed at 65˚C or 59˚C depending on the samples. The enriched solutions were then PCR-amplified for 14 to 16 cycles, after removal of the streptavidin beads. PCR products were purified using AMPure (Agencourt) with a ratio of 1.6, and paired-end sequenced on two Illumina HiSeq 2500 lines. Illumina sequencing and demultiplexing were subcontracted.

## Assembly and genotyping

For RNA-seq data (i.e. fowls and two rodents), we used trimmomatic [59] to remove Illumina adapters and reads with a quality score below 30. We constructed *de novo* transcriptome assemblies for each species following strategy B in [60], using Abyss [61] and Cap3 [62]. Open reading frames (ORFs) were predicted using the Trinity package [63]. Contigs carrying ORF shorter than 150 bp were discarded. Filtered RNA-seq reads were mapped to this assembly using Burrow Wheeler Aligner (BWA) (version 0.7.12-r1039) [64]. Contigs with a coverage across all individual below 2.5xn (where n is the number of individuals) were discarded.

Diploid genotypes were called according to the method described in [65] and [66] (model M1) via the software reads2snps (https://kimura.univ-montp2.fr/PopPhyl/index.php?section=tools). This method calculates the posterior probability of each possible genotype in a maximum likelihood framework. Genotypes supported by a posterior probability higher than 95% are retained, otherwise missing data is called. We used version of the method which accounts for between-individual, within-species contamination as introduced in [55], using the -contam = 0.1 option, which means assuming that up to 10% of the reads assigned to one specific sample may actually come from a distinct sample, and only validating genotypes robust to this source of uncertainty.

For primates, rodents, passerines and flies, reference genomes, assemblies and annotations files were downloaded from Ensembl (release 89) and NCBI (**S1 Table**). We kept only 'CDS' reports in the annotations files, corresponding to coding exons, which were annotated with the automatic Ensembl annotation pipeline, and the havana team for *Homo sapiens*. We used trimmomatic to remove Illumina adapters, to trim low-quality reads (i.e. with an average base quality below 20), and to keep only reads longer than 50bp. Reads were mapped using BWA [64] on the complete reference assembly. We filtered out hits with mapping quality below 20 and removed duplicates, and we extracted mapping hits corresponding to regions containing coding sequences according to the annotated reference assembly. This was done to avoid calling SNPs on the whole genome, which would be both time consuming and useless in the present context. We called SNPs using a pipeline based on GATK (v3.8-0-ge9d80683). Roughly, this pipeline comprised two rounds of variant calling separated by a base quality score recalibration. Variant calling was first run on every individuals from every species using Haplotype-Caller (—emitRefConfidence GVCF—genotyping_mode DISCOVERY -hets 0.001). The variant callings from all individuals of a given species were then used to produce a joint genotype using GenotypeGVCFs. Indels in the resulting vcf files were then filtered out using vcftools. The distributions of various parameters associated with SNPs were then used to set several hard thresholds (i.e. Quality by Depth < 3.0; Fisher Strand > 10; Strand Odds Ratio > 3.0; MQRootMeanSquare < 50; MQRankSum < -0.5; ReadPosRankSum < -2.0) in order to detect putative SNP-calling errors using VariantFiltration. This erroneous SNPs were then used for base quality score recalibration of the previously created mapping files using BaseRecalibrator. These mappings with re-calibrated quality scores were then used to re-call variants (HaplotypeCaller), to re-produce a joint genotype (GenotypeGVCFs,—allsites) and to re-set empirical hard thresholds (i.e. same values as above, except for Quality by Depth < 5.0). The obtained vcf files were converted to fasta files (i.e. producing two unphased allelic sequences per individual) using custom python scripts while discarding exons found on both mitochondrial and sexual chromosomes and while filtering out additional SNPs: we removed SNPs with a too high coverage (thresholds were empirically set for each species), with a too low coverage (i.e. 10x per individual) and with a too low genotype quality per individual (i.e. less than 30).

For reads generated through target capture experiment, we cleaned reads with trimmomatic to remove Illumina adapters and reads with a quality score below 30. For each species, we chose the individual with the highest coverage and constructed de novo assemblies using the same strategy as in fowls. Reads of each individuals were then mapped to the newly generated assemblies for each species, using BWA [64]. Diploid genotypes were called using the same protocol as in fowls. We used a version of the SNP calling method which accounts for between-individual, within-species contamination as introduced in [55] (see the following section). As the newly generated assemblies likely contained intronic sequences, the predicted cDNAs were compared to the reference transcriptome using blastn searches, with a threshold of e-value of 10e-15. We used an in-house script to remove any incongruent correspondence or inconsistent overlap between sequences from the transcriptomic references and the

predicted assemblies, and removed six base pairs at each extremity of the resulting predicted exonic sequences. These high-confidence exonic sequences were used for downstream analyses.

## Contamination detection and removal

For the newly generated data set, we performed two steps of contamination detection. First, we used the software tool CroCo to detect inter-specific contamination in the *de novo* assembly generated after exon capture [33].

CroCo is a database-independent tool designed to detect and remove cross-contaminations in assembled transcriptomes of distantly related species. This program classifies predicted cDNA in five categories, "clean", "dubious", "contamination", "low coverage" and "high expression".

Secondly, we used a version of the SNP calling method which accounts for between-individual, within-species contamination as introduced in [55], using the -contam = 0.1 option. This means assuming that up to 10% of the reads assigned to one specific sample may actually come from a distinct sample, and only validating genotypes robust to this source of uncertainty.

## Orthology prediction and divergence analysis

In primates, we extracted one-to-one orthology groups across the six species from the Ortho-MaM database [67, 68].

In fowls, passerines, rodents and flies, we translated the obtained CDS into proteins and predicted orthology using OrthoFinder [69]. In fowls, full coding sequences from the well-annotated chicken genome (Ensembl release 89) were added to the dataset prior to orthology prediction, then discarded. We kept only orthogroups that included all species. We aligned the orthologous sequences with MACSE (Multiple Alignment for Coding SEquences [70].

In each of earth worms, ribbon worms, mussels, butterflies and ants, orthogroups were created via a a blastn similarity search between predicted exonic sequences reference transcriptomes. In each taxon, we concatenated the predicted exonic sequences of each species that matched the same ORF from the reference transcriptome and aligned these using MACSE. We then kept alignments comprising exactly one sequence per species or if only one species was absent.

We estimated lineage specific dN/dS ratio using bppml (version 2.4) and MapNH (version 2.3.2) [71], the former for estimating each branch length and the latter for mapping substitutions on species specific branches.

Tree topologies were obtained from the literature (**S4 Table**). In passerines, fowls, rodents, flies and primates, we kept only alignments comprising all the species. In the other groups we also kept alignments comprising all species but one. We also estimated dN/dS ratios at group level by adding up substitution counts across branches of the trees, including internal branches.

To account for GC-biased gene conversion, we modified the MapNH software such that only GC-conservative substitutions were recorded [26]. We estimated the non-synonymous and synonymous number of GC-conservative sites per coding sequence using an in-house script. We could then compute the dN/dS ratio only for GC-conservative substitutions.

## Polymorphism analysis

For each taxon, we estimated ancestral sequences at each internal node of the tree with the Bio ++ program SeqAncestor [71]. The ancestral sequences at each internal node were used to

orientate single nucleotide polymorphisms (SNPs) of species that descend from this node. We computed non-synonymous ($\pi_n$) and synonymous ($\pi_s$, i.e. $\theta$) nucleotide diversity, as well as $\pi_n/\pi_s$ using the software dNdSpiNpiS_1.0 developed within the PopPhyl project (https://kimura.univ-montp2.fr/PopPhyl/index.php?section=tools) (using gapN_site = 4, gapN_seq = 0.1 and median transition/transversion ratio values estimated by bppml for each taxonomic group). We computed folded synonymous and non-synonymous site frequency spectra both using all mutations and only GC-conservative mutations using an in-house script as in [26].

## Mc-Donald-Kreitman analysis

We estimated $\alpha$, $\omega_a$ and $\omega_{na}$ using the approach of [16] as implemented in [18] (program Grapes v.1.0). It models the distribution of the fitness effects (DFE) of non-synonymous mutations, which is fitted to the synonymous and non-synonymous site frequency spectra (SFS) computed for a set of genes. This estimated DFE is then used to deduce the expected dN/dS under near-neutrality. The difference between observed and expected dN/dS provides an estimate of the proportion of adaptive non-synonymous substitutions, $\alpha$. The per mutation rate of adaptive and non-adaptive amino-acid substitution were then obtained as following: $\omega_a = \alpha$ (dN/dS) and $\omega_{na} = (1-\alpha)$(dN/dS). We computed these statistics for each species using the per branch dN/dS ratio, using either all mutations and substitutions, or only GC-conservative mutations and substitutions.

We used three different distributions to model the fitness effects of mutations that have been shown to perform the best in [18], models called GammaZero, GammaExpo and ScaledBeta in [18]. Two of these models, GammaExpo and ScaledBeta, account for the existence of segregating weakly beneficial non-synonymous mutations (i.e. beneficial mutations that contribute to the non-synonymous SFS): in GammaExpo, the positive DFE is modeled as an exponential distribution, and in ScaledBeta, the DFE for both negative and positive weakly selected mutations (with S (i.e. $4N_es$) ranging from -25 to 25) is modeled as a rescaled Beta distribution. We then averaged the estimates of the three models using Akaike weights as follows:

$$\alpha = \alpha_{GammaZero} * AICw_{GammaZero} + \alpha_{GammaExpo} * AICw_{GammaExpo} + \alpha_{ScaledBeta} * AICw_{ScaledBeta}$$

$$\omega_a = \omega_{aGammaZero} * AICw_{GammaZero} + \omega_{aGammaExpo} * AICw_{GammaExpo} + \omega_{aScaledBeta} * AICw_{ScaledBeta}$$

$$\omega_{na} = \omega_{naGammaZero} * AICw_{GammaZero} + \omega_{naGammaExpo} * AICw_{GammaExpo} + \omega_{naScaledBeta} * AICw_{ScaledBeta}$$

where AICw stands for akaike weights that were estimated using the akaike.weights function in R (https://www.rdocumentation.org/packages/qpcR/versions/1.4-1/topics/akaike.weights). Species estimates of $\alpha$, $\omega_a$ and $\omega_{na}$ for each model, as well as the associated likelihood and AIC weights are reported in **S6 Table**.

When estimating DFE model parameters, we accounted for recent demographic effects, as well as population structure and orientation errors, by using nuisance parameters, which correct each class of frequency of the synonymous and non-synonymous SFS relative to the neutral expectation in an equilibrium Wright–Fisher population [39].

We also estimated $\alpha$, $\omega_a$ and $\omega_{na}$ at group level. Two approaches were used. Firstly, we pooled species specific SFS from each group, and used the dN/dS ratio of the total tree of each taxon. We did so following the unweighted and unbiased strategy of [34], which combines polymorphism data across species with equal weights. Briefly, we divided the synonymous and non-synonymous number of SNPs of each category of the SFS of each species by the total number of SNPs of the species, then we summed those normalized numbers across species

and finally we transformed those sums so that the total number of SNPs of the pooled SFS matches the total number of SNPs across species. The resulting estimate was called $\omega_{a[P]}$. Secondly, we calculated the arithmetic mean of $\omega_{na}$ across species within a taxonomic group to obtain a non-adaptive substitution rate at the group level. We then subtracted this average from the dN/dS ratio calculating across the whole tree of each taxon to obtain an estimate of the adaptive substitution rate at group level (called $\omega_{a[A]}$).

We obtained 95% confidence intervals for species-level estimates and pooled group-level estimates by bootstrapping SNPs of the SFSs.

## Life history traits variables

Five life history traits were retrieved from the literature for each species: adult size (i.e. the average length of adults), body mass (i.e. the mean body mass of adults' wet weights), fecundity (i.e. the number of offspring released per day), longevity (i.e. the maximal recorded longevity in years), and propagule size (i.e. the size of the juvenile or egg or larva when leaving parents or group of relatives) (**S5 Table**). In the case of social insects and birds, parental care is provided to juveniles until they reach adult size so in these cases, propagule size is equal to adult size.

## Simulations

In order to evaluate whether our method to estimate the adaptive substitution rate could lead to a spurious correlation between $\pi_s$ and $\omega_a$, we simulated the evolution of coding sequences in a single population undergoing demographic fluctuations using SLIM V2 [72]. We considered panmictic populations of diploid individuals whose genomes consisted of 1500 coding sequences, each of 999 base pairs. We set the mutation rate to 2.2e-9 per base pair per generation, the recombination rate to 10e-8 per base (as in [23]) and the DFE to a gamma distribution of mean -740 and shape 0.14 for the negative part, and to an exponential distribution of mean $10^{-4}$ for the positive part (those DFE parameters correspond to the DFE estimated from the pooled SFS of primates). We simulated several demographic scenarios with four regimes of frequency of the fluctuations, as well as four regimes of intensity of the fluctuations (see **S5 Fig**). We sampled polymorphism and divergence for 20 individuals at several time points during the simulations, evaluated $\pi_s$ and $\omega_a$ and measured the correlation between the two variables.

## Supporting information

**S1 Text. Rationale of the estimation of the per group adaptive substitution rate "A".**
(PDF)

**S1 Table. Details of the species used in this study and numbers of individuals for each species.**
(DOC)

**S2 Table. Number of orthogroups for each taxonomic group.** The differences in terms of number of orthogroups comes from the fact that we not only kept orthogroups with all species but also orthogroups with all species but one to estimate dN/dS value for each terminal branches in order to maximize the number of substitutions for data sets generated by exon capture.
(XLSX)

**S3 Table. SNPs counts for each species.** SNPs counts are not integers because they corresponds to SNPs that are present in our SFS, where we chose a sample size (i.e. the number of

categories of the SFS) lower that $2*n$, where n is the number of individuals. This is to compensate the uneven coverage between individuals that results in some sites in some individuals not to be genotyped. We chose sample sizes that maximize the number of SNPs in each SFS.
(PDF)

**S4 Table. Sources of the tree topologies of each taxonomic group used to estimate branch length and map substitutions.**
(PDF)

**S5 Table. Values and sources of the life history traits used in this study.**
(DOC)

**S6 Table. Report of species estimates of life history traits, dN/dS, $\pi_s$, Tajima's D and $F_{is}$, as well as $\alpha$, $\omega_a$ and $\omega_{na}$ for each model and model averaged via AIC weights.**
(XLSX)

**S1 Fig. Cross contamination network for *de novo* assemblies from exon capture.** Circles represent the assemblies, and arrows and their corresponding numbers represent the number of cross contaminants. Most cross contamination events occur between closely-related species and are therefore likely false positive cases.
(TIF)

**S2 Fig. Relationship between $\omega_{a[P]}$ and $\pi_s$ and $\log_{10}$ transformed life history traits.** $\omega_{a[P]}$ is estimated using all mutations and substitutions (A) or using only GC-conservative mutations and substitutions (B). Group level $\pi_s$ and life history traits are estimated by averaging species level estimates across closely related species. Black dotted lines represent significant regressions across taxonomic groups and grey dotted lines non-significant ones.
(TIF)

**S3 Fig. Relationship between $\omega_{a[A]}$ and $\pi_s$ and $\log_{10}$ transformed life history traits.** $\omega_{a[A]}$ is estimated using all mutations and substitutions (A) or using only GC-conservative mutations and substitutions (B). Group level $\pi_s$ and life history traits are estimated by averaging species level estimates across closely related species. Black dotted lines represent significant regressions across taxonomic groups and grey dotted lines non-significant ones.
(TIF)

**S4 Fig. Relationship between species-level $\omega_{na}$ and $\pi_s$ and $\log_{10}$ transformed life history traits.** $\omega_{na}$ is estimated using all mutations and substitutions (A) or using only GC-conservative mutations and substitutions (B). Black dotted lines represent significant regressions across taxonomic groups and grey dotted lines non-significant ones.
(TIF)

**S5 Fig. Relationship between species-level $\alpha$ and $\pi_s$.** $\alpha$ is estimated using all mutations and substitutions (A) or using only GC-conservative mutations and substitutions (B). The dotted line represents the regression across all species, and full lines represent the regression within each taxonomic groups. Black dotted lines represent significant regressions across taxonomic groups and grey dotted lines non-significant ones.
(TIF)

**S6 Fig. Design of the simulations of fluctuation of population size.** A: three fold ratio between low and high population size and high long-term population size.
B: thirty fold ratio between low and high population size and high long-term population size.
C: three fold ratio between low and high population size and low long-term population size.

D: thirty fold ratio between low and high population size and low long-term population size.
(TIF)

**S7 Fig. Relationship between $\omega_a$ and $\pi_s$ in simulated scenarios of fluctuating population size.** A: three fold ratio between low and high population size and high long-term population size (scenario A in S1 Fig)
B: thirty fold ratio between low and high population size and high long-term population size (scenario B in S1 Fig)
C: three fold ratio between low and high population size and low long-term population size (scenario C in S1 Fig)
D: thirty fold ratio between low and high population size and low long-term population size (scenario D in S1 Fig)
(TIF)

## Acknowledgments

We thank Roger de Vila, Lise Dupont, Nicolas Bierne and Christophe Galkowski for providing us with tissue or DNA samples. We thank Iago Bonicci for the homemade program that allowed us to remove intronic sequences of the contigs obtained during the capture experiment by identifying any incongruent correspondence or inconsistent overlap on both the transcriptomic reference and the assembly of the capture experiment contigs. We also thank Yoann Anselmetti, Thibault Leroy and Jonathan Romiguier for their frequent help, Yoann Anciaux for sharing his experience with Fisher's geometrical model predictions, and Thomas Bataillon for his pieces of advice with regards to the DFE-α method models and estimates.

This preprint has been reviewed and recommended by Peer Community In Evolutionary Biology (https://doi.org/10.24072/pci.evolbiol.100080).

## Author Contributions

**Conceptualization:** Marjolaine Rousselle, Benoit Nabholz, Nicolas Galtier.

**Data curation:** Marjolaine Rousselle, Paul Simion, Marie-Ka Tilak, Emeric Figuet.

**Formal analysis:** Marjolaine Rousselle.

**Funding acquisition:** Nicolas Galtier.

**Investigation:** Marjolaine Rousselle, Nicolas Galtier.

**Methodology:** Marjolaine Rousselle, Paul Simion, Benoit Nabholz, Nicolas Galtier.

**Project administration:** Nicolas Galtier.

**Resources:** Marjolaine Rousselle.

**Software:** Marjolaine Rousselle, Paul Simion, Benoit Nabholz, Nicolas Galtier.

**Supervision:** Benoit Nabholz, Nicolas Galtier.

**Validation:** Paul Simion, Benoit Nabholz, Nicolas Galtier.

**Visualization:** Marjolaine Rousselle, Benoit Nabholz, Nicolas Galtier.

**Writing – original draft:** Marjolaine Rousselle.

**Writing – review & editing:** Marjolaine Rousselle, Paul Simion, Benoit Nabholz, Nicolas Galtier.

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
