## [Decision Letter · Decision Letter 0]

28 Jan 2020

Dear Dr Rousselle,

Thank you very much for submitting your Research Article entitled 'Is adaptation limited by mutation ? A timescale-dependent effect of genetic diversity on the adaptive substitution rate in animals.' to PLOS Genetics. Your manuscript was fully evaluated at the editorial level and by three independent peer reviewers. The reviewers appreciated the attention to an important topic but identified some aspects of the manuscript that should be improved.

We therefore ask you to modify the manuscript according to the review recommendations before we can consider your manuscript for acceptance. Your revisions should address the specific points made by each reviewer.

[LINK]

Yours sincerely,

Jianzhi Zhang

Associate Editor

PLOS Genetics

Kirsten Bomblies

Section Editor: Evolution

PLOS Genetics

Reviewer's Responses to Questions

**Comments to the Authors:**

Reviewer #1: The authors have successfully addressed all my comments and corrected most typos. The discussion is richer now than in previous versions and the results are represented and discussed more clearly. This manuscript revisits a fundamental question in population genetics and represents a major stepping stone in our field. I do not have any other major or minor comment (just check for more typos, for example line 452 end of the sentence).

Reviewer #2: In their manuscript, Rousseau et al. use a large dataset resulting from an impressive sampling effort of coding sequences in very diverse animal species, to provide an updated theory of the differences in adaptation rates between species. Even though the sequencing effort made by the authors is impressive, some of the correlations found by the authors still have limited statistical support. However, the authors convincingly present their results as supporting their proposed model of adaptation rate evolution. The authors are careful to present their results as a working theory that will require more data to validate in the future.

I do not believe that makes this paper less important, quite the opposite actually. This is a landmark paper that will guide efforts in the next five to ten years to understand differences in rates of adaptation between species. It provides at last a broadly explanatory roadmap for what to test to explain different rates of adaptation between different species, beyond the annoyingly simplistic past claims that adaptation just correlates linearly with population size.

In that respect, I share the same views as Adam Eyre Walker about the limitations of the study, but I also strongly believe that these limitations are unavoidable until we have at least an order of magnitude more coding sequence data to explore the proposed model further. This does not decrease the great merit of the manuscript, which is to pave the way for a more comprehensive understanding. Not everything can be 100% certain at first, and believing so represents a gross misunderstanding of the scientific process, that hurts and slows down scientific progress.

I have only a few comments. Castellano and Eyre Walker made great points and have covered a lot of ground already. The reviewers from PCI Evol Biol also covered a lot of ground and I will not repeat their requests as they appear to have been properly taken into account by the authors.

In the methods I could not find the part on the accounting for slightly beneficial mutations that do not fix so fast that they do not contribute to PN. This is potentially important and Galtier 2016 is a little succinct on how the MK test is robust to slightly beneficial mutations creating an excess of high frequency non-synonymous variants that can bias adaptation rate estimates. The authors need to elaborate more.

Given recent results on strong purifying selection at synonymous sites in species such a s Drosophila, I believe that another complicating factor could be the amount of purifying or positive selection at synonymous sites, and how it varies depending on population size. This can be mentioned succinctly in the Discussion.

Minor:

P14-l341: you need to specify which kind of artefact related to fluctuations in population size. I assume that the authors refer more specifically to recently smaller population sizes, that may result in both smaller Pi_s and higher PN/PS and thus lower omega_a?

Reviewer #3: I previously reviewed this manuscript for PLoS Biology and I'm happy that the authors have addressed my concerns and comments. This is a very interesting analysis and I'm happy to recommend acceptance.

Adam Eyre-Walker

**Have all data underlying the figures and results presented in the manuscript been provided?**

Reviewer #1: Yes

Reviewer #2: Yes

Reviewer #3: Yes

PLOS authors have the option to publish the peer review history of their article (what does this mean?). If published, this will include your full peer review and any attached files.

Reviewer #1: Yes: David Castellano

Reviewer #2: No

Reviewer #3: Yes: Adam Eyre-Walker

---

## [Editor Report · Decision Letter 1]

14 Feb 2020

Dear Dr Rousselle,

We are pleased to inform you that your manuscript entitled "Is adaptation limited by mutation ? A timescale-dependent effect of g enetic diversity on the adaptive substitution rate in animals." has been editorially accepted for publication in PLOS Genetics. Congratulations!

Yours sincerely,

Jianzhi Zhang

Associate Editor

PLOS Genetics

Kirsten Bomblies

Section Editor: Evolution

PLOS Genetics

Comments from the reviewers (if applicable):

**Data Deposition**

http://datadryad.org/submit?journalID=pgenetics&manu=PGENETICS-D-19-02068R1

**Press Queries**

---

## [Editor Report · Acceptance letter]

27 Mar 2020

PGENETICS-D-19-02068R1 

Is adaptation limited by mutation? A timescale-dependent effect of genetic diversity on the adaptive substitution rate in animals. 

Dear Dr Rousselle, 

We are pleased to inform you that your manuscript entitled "Is adaptation limited by mutation? A timescale-dependent effect of genetic diversity on the adaptive substitution rate in animals." has been formally accepted for publication in PLOS Genetics! Your manuscript is now with our production department and you will be notified of the publication date in due course.

With kind regards,

Jason Norris

PLOS Genetics

On behalf of:
